# Slow Sulfide Donor GYY4137 Increased the Sensitivity of Two Breast Cancer Cell Lines to Paclitaxel by Different Mechanisms

**DOI:** 10.3390/biom14060651

**Published:** 2024-05-31

**Authors:** Veronika Liskova, Barbora Chovancova, Kristina Galvankova, Ladislav Klena, Katarina Matyasova, Petr Babula, Marian Grman, Ingeborg Rezuchova, Maria Bartosova, Olga Krizanova

**Affiliations:** 1Institute of Clinical and Translational Research, Biomedical Research Center, Slovak Academy of Sciences, Dubravska Cesta 9, 845 05 Bratislava, Slovakia; veronika.liskova@savba.sk (V.L.); kristina.galvankova@savba.sk (K.G.); ladislav.klena@savba.sk (L.K.); marian.grman@savba.sk (M.G.); 2Department of Physiology, Faculty of Medicine, Masaryk University, Kamenice 753/5, 625 00 Brno, Czech Republic; babula@med.muni.cz; 3Institute of Virology, Biomedical Research Center, Slovak Academy of Sciences, Dubravska Cesta 9, 84505 Bratislava, Slovakia; viruorav@savba.sk (I.R.); maria.bartosova@savba.sk (M.B.)

**Keywords:** paclitaxel, breast cancer cell lines, apoptosis, slow sulfide donor, metabolism

## Abstract

Paclitaxel (PTX) is a chemotherapeutic agent affecting microtubule polymerization. The efficacy of PTX depends on the type of tumor, and its improvement would be beneficial in patients’ treatment. Therefore, we tested the effect of slow sulfide donor GYY4137 on paclitaxel sensitivity in two different breast cancer cell lines, MDA-MB-231, derived from a triple negative cell line, and JIMT1, which overexpresses HER2 and is resistant to trastuzumab. In JIMT1 and MDA-MB-231 cells, we compared IC50 and some metabolic (apoptosis induction, lactate/pyruvate conversion, production of reactive oxygen species, etc.), morphologic (changes in cytoskeleton), and functional (migration, angiogenesis) parameters for PTX and PTX/GYY4137, aiming to determine the mechanism of the sensitization of PTX. We observed improved sensitivity to paclitaxel in the presence of GYY4137 in both cell lines, but also some differences in apoptosis induction and pyruvate/lactate conversion between these cells. In MDA-MB-231 cells, GYY4137 increased apoptosis without affecting the IP_3_R1 protein, changing the morphology of the cytoskeleton. A mechanism of PTX sensitization by GYY4137 in JIMT1 cells is distinct from MDA-MB-231, and remains to be further elucidated. We suggest different mechanisms of action for H_2_S on the paclitaxel treatment of MDA-MB-231 and JIMT1 breast cancer cell lines.

## 1. Introduction

Breast cancer is one of the most common types of cancer affecting women, particularly in modern civilizations. Several different types of ductal or lobular breast tumors have been described up to now, differing in initiation, growth, invasiveness, treatment, and prognosis. Triple-negative breast cancer accounts for 10–20% of diagnosed breast tumors, and is considered the most aggressive subtype [1]. Other molecular subtypes of breast tumors may or may not express three important receptors: estrogen receptor (ER), progesterone receptor (PR), and human epidermal growth factor type 2 (herceptin receptor type 2; HER2).

Treatment of breast cancer differs according to the type of tumor, but generally includes chemotherapy, surgery, radiation therapy, hormonal therapy, and immunotherapy. The common chemotherapy includes cyclophosphamide, doxorubicin, methotrexate, fluorouracil, and taxanes, usually used in combination. Paclitaxel-based adjuvant chemotherapy is still recommended as a first-line regimen for treating patients with triple-negative breast cancer [2]. Various combinations of paclitaxel with different compounds were tested to improve the efficiency of this chemotherapeutic. Promising results were observed by a combination of paclitaxel with encequidar, a novel P-glycoprotein pump inhibitor [3], hyaluronic acid nanogel [4], curcumin, vitamin D3 [5], arachidin 1 [6], etc.

Paclitaxel (PTX) is a tetracyclic diterpenoid that targets microtubule polymerization. In addition to breast cancer [7], PTX is also used in the treatment of some other malignancies, e.g., ovarian cancer, pancreatic cancer, lung cancer, etc. [8]. Although PTX is not effective enough to treat colorectal carcinoma, increased sensitivity of these cells to PTX could be achieved by co-treatment with a slow sulfide donor GYY4137, probably through potentiation of apoptosis [9]. However, the mechanism of this effect is not fully clear yet.

Hydrogen sulfide (H_2_S) is a gasotransmitter that freely passes through the plasma membrane and modulates a variety of cell functions [10]. At the physiological pH, the percent distribution of H_2_S:HS^−^ is ~30:70, with traces of S^2–^ [11]. Under normal physiological conditions, it is unlikely that free endogenously produced H_2_S has a significant antioxidant role [12]. H_2_S can exert a reducing effect on the disulfide bonds of many proteins with the possible formation of protein persulfides. In cancer cells, H_2_S was shown to have a dual behavior regarding proliferation and cell death. Low concentrations of H_2_S cause cancer cell proliferation, while high concentrations are cytotoxic [13]. Therefore, many H_2_S donors have been studied for their anti-carcinogenic effect [14]. Some of the H_2_S donors were already applied for the treatment of thyroid cancer [15].

At least part of the H_2_S effect on paclitaxel could be explained by the effect of reducing agents on cancer cells. It was shown that interchain disulfide bonds of antibodies can be reduced by agents such as tris(2-carboxyethyl)phosphine (TCEP) or dithiothreitol (DTT) to form reactive cysteine residues [16]. We propose that reactive cysteines might also be formed on other proteins, and they can conjugate with paclitaxel and affect its chemotherapeutic effect. The concentration of reducing agents seems to be very important; low concentrations do not provide sufficient reducing effect, while high concentrations might be toxic. The toxicity of TCEP on human cells generating a health risk has already been described [17].

Based on the current results and also our previous study [9], we built up a hypothesis that also in breast cancer cells exogenous H_2_S can increase sensitivity to paclitaxel. Breast cancers form a wide group of tumors which differ in the mechanism of origin, invasiveness, and prognosis. Nevertheless, they are more sensitive to taxanes compared to colorectal cancer. The effect of paclitaxel might depend on the type of breast cancer tumor. Thus, this work aims to compare the combined effect of paclitaxel and a slow sulfide donor GYY4137 as a reducing agent on triple negative MDA-MB-231 cells and HER2-positive JIMT1 breast cancer cells and try to elucidate the mechanism of possible effect observed. We have chosen two different breast cancer cell lines to elucidate the effect of GYY4137 on paclitaxel treatment. A JIMT1 cell line was established from the pleural effusion of a 62-year-old woman with ductal breast cancer. JIMT1 carries an amplified HER2 oncogene and is insensitive to HER2-inhibiting drugs, e.g., trastuzumab [18]. The MDA-MB-231 cell line is an epithelial human breast cancer cell line that was established from a pleural effusion of a 51-year-old Caucasian female with a metastatic mammary adenocarcinoma. This cell line represents a highly aggressive, invasive, and poorly differentiated triple-negative breast cancer cell line, as it lacks estrogen receptor (ER) and progesterone receptor (PR) expression, as well as HER2 (human epidermal growth factor receptor 2) amplification. Because of the different origins of both cell lines, we expect some differences in biochemical and/or physiological parameters, since MDA-MB-231 cells are more aggressive and less responsive to PTX treatment.

## 2. Materials and Methods

### 2.1. Cell Cultivation and Treatments

Two breast cancer cell lines were used for all experiments: the MDA-MB-231 cell line (HTB-26, ATCC, Manassas, VA, USA), and the JIMT1 cell line (ACC 589, DSMZ, Braunschweig, Germany). Both cell lines were authenticated in Testing Laboratory no. 1650 accredited ČIA according to ČSN EN ISO/IEC 17025:2018, case marketing 3397-7/2020 and 3397-1/2020. Cell lines were cultured in Dulbecco Minimal Essential Medium (DMEM; Sigma-Aldrich; Burlington, MA, USA) with high glucose (4.5 g/L) and L-glutamine (300 μg/mL), supplemented with 10% fetal bovine serum (FBS; Sigma-Aldrich) and penicillin/streptomycin mixture (Calbiochem, San Diego, CA, USA; penicillin 100 U/mL; streptomycin 100 μg/mL). Cells were cultured in a humid atmosphere at 37 °C and 5% CO_2_ [19]. Experiments were performed within the cell passages 5–15, and they were tested weekly for mycoplasma using double-nested PCR. Some groups of cells were treated with paclitaxel (PTX; Selleckchem, Pittsburgh, PA, USA; 2 nM, 20 nM, 200 nM) or slow-releasing sulfide donor GYY4137 (GYY; Cayman Chemical, Ann Arbor, MI, USA; 100 µM) for 24 h. The optimal concentration of GYY4137 was determined previously in our work [9]. As a control of GYY4137, dithiothreitol (DTT; Sigma-Aldrich; 1 mM) and tris(2-carboxyethyl) phosphine (TCEP; Sigma-Aldrich; 1 mM) were used. The optimal concentration of DTT was determined using a concentration-dependent annexin V assay, and concentration of TCEP was chosen based on the results from Ta et al. [20], since a higher concentration of TCEP was reported to affect the expression of tubulins.

### 2.2. Cell Viability Determination Using AquaBluer and IC50 Determination

The assay was carried out in accordance with the manufacturer’s recommendations (AquaBluer Solution MultiTarget Pharmaceuticals LLC, Colorado Springs, CO, USA). Briefly, cells in concentration 7 × 10^3^ were plated on a 96-well plate 24 h prior to the treatments [9]. After 24 h treatment, 100 µL diluted AquaBluerTM (1:100) was added. After 4 h incubation at 37 °C and 5% CO_2_ in the dark, fluorescence was measured on the fluorescence scanner Synergy II (Biotek, Bad Friedrichshall, GE) at λex 540 nm and λem 590 nm. Viability was calculated with the formula %Viability = (RFU test/RFU veh) × 100, and IC50 was calculated using CalcuSyn software version 1.1 (Biosoft, 1996).

### 2.3. Apoptosis Determination Using Annexin V-FLUOS

Determination of apoptosis was performed in 24-well plates, as described in Kajsik et al. [9]. After 24 h treatment, cells were gently removed using trypsin and pelleted at 1000× *g* for 5 min. Cells were washed with 1 mL of phosphate saline buffer (PBS), and the cell pellet was resuspended in 50 μL of Annexin V Binding Buffer (BioVision, San Francisco, CA, USA) with 2 µL of Annexin-V-FLUOS (Roche Diagnostics, Indianapolis, IN, USA) and incubated at room temperature in the dark for 20 min according to the manufacturer’s protocol. After the incubation, 200 µL of Annexin V Binding Buffer and 5 µL of 7-AAD Viability Staining Solution (7-amino-actinomycin D, Thermo Fisher Scientific, Waltham, MA, USA) were added, samples were placed on ice, and were measured using a CytoFLEX S flow cytometer (Beckman Coulter, Brea, CA, USA). Results were analyzed using Cytexpert software version 2.4.0.28 (Beckman Coulter, Brea, CA, USA).

### 2.4. Cell Migration Assay

One million cells were plated on 3 cm Petri dishes and left to adhere for 24 h. Confluent monolayers were scratched with a pipette tip, washed twice with PBS, and supplemented with fresh culture medium without FBS or fresh culture medium with GYY4137, PTX, or their combination without FBS. Images were taken after 24 h. Afterwards, the medium was replaced with fresh culture medium supplemented with 10% FBS, and cells were grown for another 24 h before images were taken thereafter. Cell migration was evaluated as % of an overgrown wound and expressed as means ± S.E.M. Measurements were repeated at least 8 times from different cultivations.

### 2.5. MTT Assay

Cell proliferation was determined using the MTT assay. The MTT Assay Protocol from Abcam was used with some modifications (https://www.abcam.com/en-sk/technical-resources/protocols/mtt-assay), accessed on 13 May 2024. Briefly, MDA-MB231 and JIMT1 cells were seeded in a 96-well plate at a density of 10,000 cells per well for each condition—PTX (20 nm), GYY41379 (100 µm), PTX (20 nm) + GYY4137 (100 µm), and untreated controls. Cells for each condition were seeded in quintuplicate. Cells were incubated for 24 h, treated, and incubated for another 24 h or 48 h. All treatments were performed in serum-free DMEM. The MTT assay was conducted after each time point using 1 mg/mL MTT in 1X PBS. Each plate was incubated with MTT + serum–free DMEM for 3 h at 37 °C, 5% CO_2_. Formazan crystals were dissolved with pre-warmed DMSO and measured at 590 nm.

### 2.6. Intracellular pH (pHi) Measurement

Cells in a concentration of 2 × 10^5^ were plated on a 24-well plate. After 24 h treatment, cells were loaded with 5 µM BCECF AM (Sigma-Aldrich, USA) and serum-free medium at 37 °C for 1 h and 5% CO_2_ in the dark, as described in Pastorek et al. [21]. Afterwards, cells were washed twice with 500 µL of PBS and fluorescence was measured at λex 490/440 nm and λem 535 on the fluorescence scanner Synergy II (Biotek, Bad Friedrichshall, GE). ΔpHi was calculated from intensity values acquired at 490/535 nm and 440/535 nm compared to the control group of cells.

### 2.7. Determination of Oxidative Stress

Cellular oxidative stress was determined by measuring reactive oxygen species (ROS) using CellROX^TM^ Orange Reagent (Thermo Fisher Scientific, Waltham, MA, USA), as described previously [22]. As a positive control, pyocyanine was used (50 μM for 4 h; Sigma-Aldrich, USA). Cells were plated on a 24-well plate at a density of 1 × 10^4^. After treatment, cells were washed with serum-free medium and incubated with CellROX^TM^ Orange Reagent at a final concentration of 5 μM per well for 30 min. at 37 °C and 5% CO_2_ in the dark. Next, cells were washed with 500 μL of PBS, and fluorescence was measured using the Synergy II Reader (Biotek, Bad Friedrichshall, GE) at λex 545 nm and λem 585 nm. The results were expressed as arbitrary units of fluorescence intensity.

### 2.8. Pyruvate Assay

Pyruvate levels were determined using the Pyruvate assay kit (BioVision, San Francisco, CA, USA), according to the manufacturer’s instructions. Absorbance was determined using a wavelength of 570 nm in a Synergy II Reader (Biotek, Bad Friedrichshall, GE). Results were expressed as nmol/50 µL of the sample.

### 2.9. Lactate Assay

Lactate levels were determined using the Lactate Assay Kit (ab65330, Abcam, Cambridge, UK), according to the manufacturer’s instructions. Absorbance was determined at a wavelength of 570 nm using the Synergy H1 Hybrid Multi-Mode Reader (Biotek, Bad Friedrichshall, GE). Results were expressed as nmol/10 µL of the sample.

### 2.10. Western Blot Analysis

Western blot analysis was performed as described in Chovancova et al. [22]. Briefly, cells were scraped and resuspended in 10 mM Tris–HCl pH 7.5, 1 mM phenylmethylsulfonyl fluoride (Sigma-Aldrich, USA) and subjected to centrifugation for 10 min at 10,000× *g* at 4 °C. The pellet was resuspended in Tris buffer containing 50 µM 3-[(3-cholamidopropyl)dimethylammonio]-1-propane sulfonate hydrate (CHAPS; Sigma-Aldrich, USA), incubated for 20 min at 4 °C and centrifuged. Protein concentration was determined using a modified Lowry Protein Assay Kit (Thermo Scientific, USA). Protein extract from each sample was separated using electrophoresis on 4–20% gradient SDS polyacrylamide gels, and proteins were transferred to Hybond PVDF blotting membrane (GE Healthcare, Life Sciences, Piscataway, NJ, USA) using semidry blotting (Owl, Inc., Kamakura, Japan). Afterwards, membranes were blocked in 5% non-fat dry milk in TBS-T (Tris-buffered Saline with Tween-20) overnight at 4 °C and then incubated for 1 h with primary antibody against beta-tubulin (1:2000, ab231082, Abcam, Cambridge, UK; 49 kDa), beta-actin (1:5000 dilution, ab6276, Abcam, Cambridge, UK; 42 kDa), and GAPDH (1:5000 dilution, ab8245, Abcam, Cambridge, UK; 36 kDa), or membranes were blocked in 5% non-fat dry milk in TBS-T for 1 h at room temperature and then incubated overnight at 4 °C with primary antibody against IP_3_R1 (1:1000 dilution, 07-1210, Merck, Darmstadt, GE; 260 kDa) and IP_3_R1 (1:1000, I157, Sigma Aldrich, USA, 240 kDa). Membranes were incubated with horse-radish peroxidase-linked secondary goat anti-mouse/anti-rabbit antibody (1:10,000 dilution, ab6789; ab205718, Abcam, UK) 1 h at room temperature and chemiluminescence detection system (LuminataTM Crescendo Western HRP Substrate, Millipore) was used for visualization. Each membrane was digitally captured using an imaging system (C-DiGit, LI-COR). The intensity of bands was determined as optical density/mm^2^.

### 2.11. Immunofluorescence

This method was already described by Chovancova et al. [22]. Cells grown on glass coverslips were fixed in ice-cold methanol. Non-specific binding was blocked using incubation with PBS containing 3% bovine serum albumin (BSA) for 60 min for 1h at room temperature. Cells were then incubated with primary antibodies diluted in PBS with 1% BSA (PBS-BSA) for 1 h at 37 °C. In these experiments, an anti-beta tubulin antibody (1:1000 dilution, ab231082, Abcam, Cambridge, UK) was used. Afterwards, cells were washed four times with PBS with 0.02% TWEEN (Sigma Aldrich, USA) for 10 min, incubated with Alexa Fluor-488 goat anti-mouse IgG (1:1000 dilution, Thermo Fisher Scientific, Waltham, USA) in PBS-BSA for 1 h at 37 °C, and washed as described previously. Finally, coverslips were mounted onto slides in mounting medium with blue fluorescent DNA stain 4′,6-diamidino-2-phenylindole (DAPI, Sigma Aldrich, USA). Images of all samples were acquired with the same microscope setup. Cells were visualized with epifluorescence microscopy using a Nikon Eclipse Ti-S/L100 (Nikon, Tokyo, Japan); NIS elements software (Nikon, Japan) was used to process images and evaluate the resultant pictures. Laser scanning confocal microscope Zeiss LSM 880 with an AiryscanFast module was used to visualize cytoskeleton details.

### 2.12. Determination of Angiogenesis Using CAM

Fertilized Japanese quail (Coturnix japonica) eggs were kindly provided by the Institute of Animal Biochemistry and Genetics, Centre of Biosciences.

Slovak Academy of Sciences. Fertilized eggs were incubated in a draught incubator at 37 °C and 60% humidity. After 56 h of development, the embryos were tipped from the eggs into six-well tissue culture plates and incubated at 37 °C and 55–60% humidity for another four days until the chorioallantoic membrane (CAM) covered the entire surface of the well. Two and a half million MDA-MB-231 or JIMT1 cells in 50 µL of PBS were plated onto the area of CAM defined by the silicone ring. Twenty-four hours after inoculation, tumor cells growing on CAM were treated with PTX, GYY4137, or their combination (in 50 µL of PBS) for 24 h. Controls were treated with 50 µL PBS without drugs. Tumors and the vasculature of the CAM were photographed before and after treatment with a Canon EOS 5D Mark II (Canon Inc., Tokyo, Japan) camera. Each group was evaluated on 10 embryos. The experiment was performed twice. Changes in the structure of the CAM vasculature were evaluated by two investigators independently, according to the following classification score: 0 = unchanged blood vessels; 1 = moderate changes: <25% of vessels are irregular in size, and shape with richly branched microcapillaries; 2 = focal changes: ≤50% of vessels are irregular in size and shape, with richly branched microcapillaries and with visible anastomoses; 3 = multifocal changes: ≤75% of vessels are irregular in size and shape, with richly branched microcapillaries and with visible anastomoses and/or microhemorrhages; and 4 = severe changes: >75% of vessels are irregular in size and shape, with richly branched microcapillaries and with visible anastomoses and microhemorrhages. The overall result in the groups—control, PTX, GYY4137, and PTX + GYY4137—was evaluated as the average score of individual cases.

### 2.13. Statistical Significance

The results are presented as mean ± S.E.M. Each value represents an average of at least 3 replicates from at least 3 independent cultivations of each type of cell. Statistical differences among groups were determined using one-way ANOVA. For multiple comparisons, an adjusted *t*-test with *p* values corrected using the Bonferroni method was used (InStat, GraphPad Prism Software, Version 3.10). Statistical significance * *p* < 0.05, ** < 0.01, and *** *p* < 0.001 were considered to be significant.

## 3. Results

To determine the effectivity of PTX on viability of the cells, we determine IC50 in each cell line. We have found that the IC50 of paclitaxel was approximately ten times higher in MDA-MB-231 cells compared to JIMT1 cells (Figure 1A). Slow sulfide donor GYY4137 (GYY) significantly decreased the IC50 of PTX in both these cell lines (Figure 1A). Treatment with GYY4137 alone was not toxic for either MDA-MB-231 (IC50-66 mmol/L) or for the JIMT1 cell line (IC50-34.5 mmol/L). We propose that apoptosis might be involved in the effect of PTX treatment. PTX increased apoptosis in a concentration-dependent manner in both MDA-MB-231 and JIMT1 cells (Figure 1B). GYY4137 further increased apoptosis induction in the MDA-MB-231 cell line, but not in the JIMT1 cell line (Figure 1B). Since at least part of the H_2_S effect on paclitaxel could be explained by its effect as a reducing agent on cancer cells, we have chosen two reducing agents—DTT and TCEP—and tried to compare their effect on apoptosis induction with the effect of GYY4137. Co-treatment of PTX with TCEP increased apoptosis in MDA-MB-231 and also in JIMT1 cells (Figure 1C). Nevertheless, necrosis in DTT- and TCEP-treated cells co-treated with PTX was much higher compared to other groups, especially in JIMT1 cells (4.6 ±0.3% in controls, 6.7 ± 1.2% in PTX/GYY4137, 9.3 ± 0.8% in PTX/DTT, and 8.9 ± 1.4% in PTX/TCEP), which might point to higher cytotoxicity of these compounds in combination with PTX. Since IP_3_R1 is known to activate the inner mitochondrial pathway of apoptosis, we also determined an expression of this transport system. IP_3_R1 protein was not changed either in MDA-MB-231 or in JIMT1 cells due to PTX and GYY4137 treatment, thus suggesting either changes in the activity of the IP_3_R1, or another mechanism of apoptosis induction (Figure 1D).

Furthermore, we proposed that changes in the intracellular pH could be due to the conversion of pyruvate to lactate. Therefore, we determined the intracellular pH (pHi) in the control and treated cells (Figure 2A,B). PTX had no significant effect on intracellular pH compared to control in MDA-MB-231, but significantly decreased levels of pHi in JIMT1 cells. A significant decrease was observed in MDA-MB-231 cells (but not in JIMT1 cells) due to GYY4137 treatment (Figure 2A,B). A rapid decrease in intracellular pH was observed in MDA-MB-231 and also the JIMT1 cell line, when cells were treated with the combination of both GYY4137 and PTX (Figure 2A,B). Therefore, levels of pyruvate (Figure 2C,D) and lactate (Figure 2E,F) were also determined in all groups of MDA-MB-231 and JIMT1 cells. As expected, in MDA-MB-231 cells, levels of pyruvate were decreased in all treated groups, with the highest drop in PTX/GYY4137 group (Figure 2C), which corresponds to the increase in lactate levels (Figure 2E). Surprisingly, in JIMT1 cells, PTX treatment did not change the levels of pyruvate, but GYY4137 significantly increased pyruvate levels either alone or in combination with PTX (Figure 2D). In groups of JIMT1 cells treated with PTX, GYY4137, or PTX/GYY4137, a rapid decrease in lactate levels was detectable compared to the control group (Figure 2F).

ROS are highly reactive molecules known to regulate several signaling pathways, e.g., they are able to trigger programmed cell death. ROS levels were significantly increased in the JIMT1 cell line, and also in the MDA-MB-231 cell line due to PTX and/or PTX/GYY4137 treatments (Figure 3A,B). A decrease in ROS levels was observed in JIMT1 cells after GYY4137 treatment, while in MDA-MB-231 cells, increased ROS levels were detected after the treatment with GYY4137. Significant changes in ROS levels were observed between GYY4137 and PTX/GYY4137 groups in both cell lines (Figure 3A,B).

PTX is known to block the depolarization of tubulin. Therefore, we detected levels of β-tubulin in MDA-MB-231 and JIMT1 cells using immunofluorescence and Western blot analysis (Figure 3C–E and Figure 4). PTX and PTX/GYY4137 showed significant elevation of β-tubulin levels, as determined using Western blot analysis in MDA-MB-231 (Figure 3C,E), but not in JIMT1 (Figure 3D,E). However, immunofluorescence using anti-β-tubulin antibody revealed a collapse of the cytoskeleton in JIMT1 cells (Figure 4A). In MDA-MB-231 cells, marked differences in the tubulin signal in PTX and PTX/GYY groups were determined—a dashed pattern in PTX-treated cells, suggesting longer polymerized b-tubulin filaments—and a dotted pattern in PTX/GYY4137-treated cells, similar to the pattern of the control sample, suggesting at least a partial depolarization of tubulin (Figure 4B).

PTX and PTX/GYY4137 can also affect processes of migration and proliferation. Changes in migration due to PTX and PTX/GYY4137 treatment were determined using a wound healing assay. The migration of the MDA-MB-231 cells was more rapid than the migration of the JIMT1 cells (Figure 5). The size of the wound was determined immediately after the scratch and after 24 and 48 h (Figure 5A,D). In both MDA-MB-231 (Figure 5A,B) and JIMT1 (Figure 5D,E) cells, the migration was significantly slower in groups treated with PTX or a combination of PTX and GYY4137, compared to GYY4137 alone. Migration in the presence of GYY4137 was similar to control untreated cells (Figure 5). Also, in both these cell lines, proliferation as a marker of cytotoxicity was determined after 24 and 48 h of the treatment (Figure 5C,F). Proliferation was lower after the PTX treatment in both cell lines, and this decrease was potentiated after combined treatment of PTX and GYY4137.

Paclitaxel also possesses an antiangiogenic property, as shown in highly vascularized transgenic murine breast cancer [23]. We evaluated the effect of PTX, GYY4137, or a combination of PTX and GYY4137 on the angiogenesis of MDA-MB-231 and JIMT1 cell line tumors on the CAM (Figure 6). Tumors growing on CAM release proangiogenic factors into the tumor microenvironment, which induce abnormal blood vessel growth. Therefore, CAM with untreated cells has a dense and irregular vascular network that surrounds and perfuses the tumor. MDA-MB-231 tumor cells induced vasculature changes, scored as 2.23 ± 0.20 and JIMT1 cells as 1.86 ± 0.31. Treatment of tumor cells causes normalization of the vascular network—atrophy of thin blood vessels that previously supplied the tumor. In the case of MDA-MB-231 cells, the combination of PTX and GYY4137 had the same effect on the normalization of the vasculature (achieved score 1.85 ± 0.18) as that of PTX treatment alone (score 1.90 ± 0.27; Figure 6A,D). We observed more rapid normalization of the CAM vasculature when JIMT1 tumors were treated with PTX (score 1.40 ± 0.37), GYY4137 (score 1.33 ± 0.21), and a combination of PTX and GYY4137 (score 1.57 ± 0.3; Figure 6B,D). However, the combined treatment of PTX with GYY4137 had the greatest effect on reducing the size and thickness of the tumor mass on the CAM. Treatment with PTX and GYY alone or in combination did not affect the tumor cell-free CAM vasculature (Figure 6C). Therefore, all the described changes in the CAM vasculature were caused by the growth of tumor cells, as well as the effect of their treatment.

## 4. Discussion

Breast cancer is a group of diseases with different origins, etiology, progression, and treatment [24,25,26]. Different combinations of chemotherapeutics are effective for individual subtypes. Nevertheless, taxanes are an important part of the treatment of all types of breast cancer. In this work, we aimed to compare the effect of paclitaxel on two different types of breast cancer cells and to elucidate the potential reducing effect of H_2_S supplied by GYY4137 (exogenous slow-releasing donor of H_2_S) on paclitaxel sensitivity to these types of cancer cells. This goal was motivated by our previous observation that in colorectal carcinoma cells, the effect of paclitaxel was increased by a slow-releasing sulfide donor GYY4137 [9]. Also, the effect of another H_2_S-releasing donor (HA-ADT) was evaluated on breast cancer cells, and this donor was able to suppress the growth and proliferation of these cells [27]. In contrast to colorectal carcinoma, taxanes, namely paclitaxel or docetaxel, have been approved as first-line therapies against all types of breast cancer [2]. We compared the effect of the PTX and PTX in combination with a slow sulfide donor GYY4137 on two breast cancer cell lines, MDA-MB-231 and JIMT1. Choice of these lines was motivated by the characteristics of tumors they were derived from. The published 50% inhibitory concentration (IC50) values determined using a cell viability assay for breast cancer cells differ significantly, e.g., in MDA-MB-231 from 2.4–5 nM [28,29] to 300 nM [30]. As we expected, the IC50 of paclitaxel in MDA-MB-231 cells was approximately ten times higher than that in JIMT1 cells. Nevertheless, co-treatment with GYY4137 decreased IC50 of paclitaxel in both these cell lines, while IC50 for GYY4137 alone was in the millimolar range in both MDA-MB-231 and JIMT1 cells. Also, in our previous work performed on colorectal cancer cells, IC50 of PTX was in the nanomolar range, and IC50 of GYY4137 was in the micromolar range, showing a low toxicity of GYY4137 after 24h treatment [9]. Since GYY4137 and other sulfide donors were already shown to potentiate apoptosis in cancer cells [31,32,33], we first evaluated the effect of GYY4137 on the level of apoptosis. Co-treatment of GYY4137 with PTX potentiated apoptosis in MDA-MB-231 cells, but not in JIMT1 cells, thus pointing to the diverse effect of exogenous H_2_S on different types of breast cancer cells. Apoptosis was not significantly affected by PTX and/or GYY4137 in non-cancer EAHy926 cells [9]. Apoptosis can be induced by different mechanisms using various stimuli, e.g., ROS, activation of IP_3_R1, reducing agents, etc. To ensure whether this potentiation is due to the reducing properties of H_2_S, we used two other reducing agents (DTT and TCEP), by which we observed a similar effect to GYY4137 in MDA-MB-231 cells, thus suggesting a reduction in disulfide bonds with H_2_S released by GYY4137 in the mechanism of increased apoptosis in MDA-MB-231 cells. The effect of reducing agents depends also on their concentration, so in JIMT1 cells we observed an increase in PTX only in combination with TCEP. TCEP was shown to induce caspase-dependent apoptosis, probably through a decrease in the expression of anti-apoptotic Bcl-XL protein in canine and human osteosarcoma cell lines [34]. ROS are highly reactive molecules able to regulate a variety of signaling pathways. The production of ROS is elevated in tumor cells, thus contributing to several pathologic conditions, e.g., tumor promotion and progression. However, ROS are also able to trigger programmed cell death [35]. PTX increases the ROS levels in both cell lines, while after the GYY4137 treatment, an increase was detected only in MDA-MB-231 cells. This observation supports the hypothesis that in MDA-MB-231 cells (but not in JIMT1 cells), apoptosis is the main executor in GYY4137-treated cells with/without PTX. IP_3_R1 participates in the induction of the inner mitochondrial pathway of apoptosis [19,36]. Expression of the IP_3_R1 protein in MDA-MB-231 cells and JIMT1 cells was not changed by any previously described treatment, which would suggest a different mechanism of apoptosis induction than through the upregulation of the IP_3_R1 protein, e.g., through increased activity of these receptors.

Dysregulation in glucose uptake and increased lactate production is a common hallmark of many solid tumors [37]. We observed differences in pyruvate and lactate levels in MDA-MB-231 compared to JIMT1 cells. Decrease in pyruvate and increase in lactate levels in MDA-MB-231 cells due to PTX, GYY4137, and combined treatment point to the mechanism of intracellular acidification. In JIMT1 cells, PTX did not affect pyruvate levels, but GYY4137 without/with PTX significantly increased these levels and decreased lactate. The intracellular pH in JIMT1 cells was decreased, which was very unusual. Since the application of PTX caused significant changes in intracellular pH, and β-tubulin levels were not increased by this treatment, we might propose that the mechanism by which PTX acts in JIMT1 cells is not only on tubulin polymerization, but may include also other, possibly energetic factors, e.g., disulfidoptosis [38]. Nevertheless, further studies are needed to uncover this mechanism. Increased levels of pyruvate in JIMT1 cells due to GYY4137 treatment might be attributed to the reduction in mitochondrial import of pyruvate due to low expression of mitochondrial pyruvate carrier 1, as it was described in colorectal cancer cells [39]. GYY4137, and thus the H_2_S, seems to also be an important modulator of pyruvate/lactate levels in cancer cells. GYY4137 decreased the level of pyruvate and increased the level of lactate in MDA-MB-231 cells. This observation is in line with Untereiner and co-workers [40], who have shown that in colorectal carcinoma HCT116 cells, exogenous H_2_S induces the stimulation of lactate dehydrogenase A, which catalyzes the conversion of pyruvate to lactate. This effect was more pronounced when a combination of PTX and GYY4137 was used. In JIMT1 cells, we propose the effect of GYY4137 with/without PTX also on β-actin, collapse of the cytoskeleton due to disulfidoptosis, and subsequent cell death.

A key marker for cellular apoptosis induction is ROS accumulation; this stimulates the ROS-dependent apoptosis signaling pathways. ROS are derived from molecular oxygen mainly by the electron transfer chain in complexes I and III, resulting in the formation of a superoxide anion radical, and subsequently hydrogen peroxide, either spontaneously or by the action of superoxide dismutase; see review [41]. In the presence of iron, superoxide and H_2_O_2_ can lead to the formation of highly reactive hydroxyl radicals, which can damage cellular proteins, RNA, DNA, and lipids. In both MDA-MB-231 and JIMT1 cells, we observe a significant increase in ROS when cells were treated with PTX and/or PTX and GYY4137. An increase in ROS due to the paclitaxel treatment was observed in prostate cancer cells [42]. We propose that an increase in ROS might affect apoptosis induction specifically in cancer cells, since in our previous studies [9] we have shown that paclitaxel treatment did not affect apoptosis induction in non-cancerous EAHy926 cells. Contrary to MBA-MB-231 cells, in JIMT1 cells, a decrease in ROS levels occurred due to GYY4137 treatment. Several studies have shown that H_2_S can readily scavenge ROS at higher rates than other classic antioxidants [43,44,45]. Exogenous H_2_S has demonstrated profound antioxidant and cytoprotective capabilities in physiologic systems exposed to ROS [46]. On the other hand, several compounds (C8-ceramide, *piper nigrum* ethanolic extract) were shown to cause overproduction of ROS that results in apoptosis [47,48]. While this explanation fully copes with our observation on JIMT1 cells, the mechanism of ROS impact on apoptosis remains to be elucidated. In MDA-MB-231 cells, GYY4137 treatment resulted in increased levels of ROS.

PTX shifts the equilibrium between soluble tubulin and the microtubule polymer in favor of the polymer, and such microtubules are resistant to depolymerization [49]. We observed increased levels of β-tubulin in MDA-MB-231, but not in JIMT1 cells due to PTX and/or PTX/GYY4137 treatment, as determined using Western blot analysis. Nevertheless, we did not find a lot of information on the effect of H_2_S on the structure of the cytoskeleton. In plants, it was described that hydrogen sulfide disturbs actin polymerization via persulfidation [50]. Based on the results from a confocal microscope, where we have seen marked differences in β-tubulin structure, we might speculate that the tubulin structure is also affected by persulfidation in the PTX/GYY4137-treated group of MDA-MB-231 cells. Although this proposal remains to be elucidated, the abovementioned explanation is highly probable, because it was already published that about 10–25% of many liver proteins, including actin and tubulin, are persulfidated [51]. In JIMT1 cells, the cytoskeleton seems to be collapsed due to PTX and was concentrated mainly around the nuclei, and GYY4137 treatment partially renewed the cytoskeleton.

Migration is the basic feature of cancer invasiveness. MDA-MB-231 cells are more vulnerable to migration than JIMT1 cells; under control conditions, they migrate more rapidly. PTX affected the migration of both types of breast cancer cells; it decelerates the migration of cells. Furthermore, we tested the effect of GYY4137 administration in the presence/absence of PTX on these cells. The effect of H_2_S on cell migration is not yet described sufficiently, and probably depends on the type of cells. It was shown that H_2_S promoted alveolar type II cell migration [52]. On the other side, rat neonatal cardiac fibroblast migration was inhibited by the administration of GYY4137 [53]. Exogenous H_2_S inhibited colon cancer cell proliferation and migration in vitro [54]. In three breast cancer cell lines (MCF7, MCF10A, and MDA-MB-231), Dong et al. [27] observed slight inhibition of migration after 24 h. In our experiments, GYY4137 did not significantly affect migration in any tested cell line. Nevertheless, the inhibitory effect of PTX on cell migration was not affected by co-treatment with GYY4137, which clearly shows that while PTX has a significant antimigratory effect, GYY4137 did not affect migration.

H_2_S has already been described to stimulate angiogenesis in various preclinical models [55,56,57]. Therefore, Macabrey et al. [58] used the hydrogen sulfide donor, or more precisely, the donor of sulfane sulfur (So), sodium thiosulfate, to potentiate neovascularization in wild-type and hypercholesterolemic LDLR−/− mice with subsequent hindlimb ischemia. Sodium thiosulfate affects migration and proliferation in a glycolysis-dependent manner. The grafting of tumors onto the CAM allows for studying the morphological aspects of the tumor and the blood vessel interactions. The most common mechanism involved in organ graft revascularization includes the formation of peripheral anastomoses, formed between a host and pre-existing donor vessels. This mechanism is the most commonly used in graft revascularization on CAM, whereas the sprouting of CAM-derived vessels into the transplants only occurs in the grafts of tumor tissue [59,60]. In our experiments, we observed increased neoangiogenesis on chorioallantoic membranes of quail embryos with engrafted MDA-MB-231 and JIMT1 cells. The new vascular network induced by the tumor cells consisted of thin, richly branched, and tortuous capillaries that rarely produced small hemorrhages. However, this was partially normalized, especially after the treatment of MDA-MB-231 cells with a combination of PTX and GYY4137.

## 5. Conclusions

In summary, we have shown that a slow-releasing sulfide donor of H_2_S can potentiate the effect of paclitaxel in two breast cancer cell lines, MDA-MB-231 and JIMT1, which represent two diverse forms of breast tumors differing in etiology, invasiveness, and also prognosis. Nevertheless, the mechanism of effect is probably different. In MDA-MB-231 cells, GYY4137 increased apoptosis, either through increased IP_3_R1 activity (and not increased protein levels), or by another mechanism, and also increased ROS production, either alone or in combination with PTX. Moreover, GYY4137 in combination with PTX significantly increases lactate production, thus resulting in the acidification of intracellular space that might also contribute to cell death. On the other hand, GYY4137 in JIMT1 cells increases the production of pyruvate while decreasing lactate and ROS production. Interestingly, in JIMT1 cells, the cytoskeleton (determined by immunostaining with β-tubulin) seems to be collapsed. Collapse of the cytoskeleton is a characteristic feature of a newly described form of cell death called disulfidoptosis (for review, see [38]). Disulfidoptosis results in the depletion of NADPH and the accumulation of intracellular disulfide molecules and cell death. Depletion of NADPH might be responsible for decreased ROS levels in the GYY4137-treated JIMT1 cells, and probably also for increased pyruvate levels in these cells, especially in GYY4137-treated cells. However, both these hypotheses remain a basis for further research in H_2_S signaling in breast cancer.

## Figures and Tables

**Figure 1 biomolecules-14-00651-f001:**
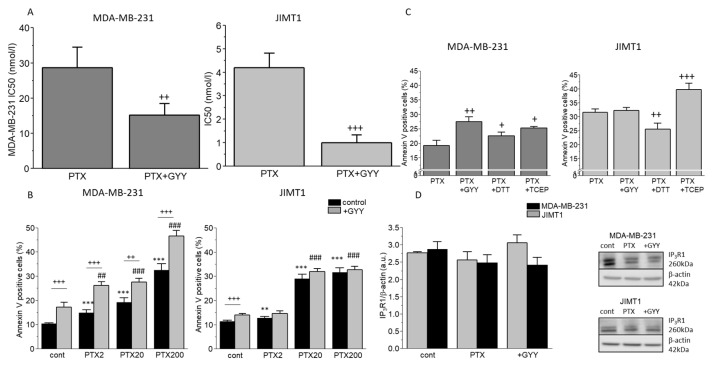
IC50 values and apoptosis after PTX and GYY4137 treatment in MDA-MB-231 (upper lane) and JIMT1 cells (lower lane). IC50 values (**A**), apoptosis levels (**B**,**C**), and protein expression of the IP_3_R1 (**D**) in two breast cancer cell lines were determined as described in the Materials and Methods Section. The IC50 values of paclitaxel (PTX) and the combination of PTX and GYY4137 (GYY) are significantly lower in JIMT1 compared to MDA-MB-231 cells (**A**). Different concentrations of PTX (2 nM, 20 nM, and 200 nM) were used with/without GYY4137 (100 µM) to measure apoptosis (**B**). We observed a concentration-dependent increase in apoptosis in both MDA-MB-231 and JIMT1 cells after PTX treatment (**B**). The combination of GYY and PTX resulted in a further increase in apoptosis in MDA-MB-231, but not in JIMT1 cells (**B**). For both cell lines, we chose the same PTX 20 nM concentration. We observed increased levels of apoptosis after PTX treatment with TCEP in both cell lines, and increased levels after PTX with GYY only in MDA-MB-231 cells (**C**). Protein expression of the IP_3_R1 was not changed in MDA-MB-231 and/or JIMT1 cells (**D**). Representative gel images are included. Results are displayed as mean ± S.E.M. and represent an average of five different cultivations, each cultivation was performed in octaplicates for IC50 values, an average of n = 7–28 parallels for apoptosis levels, and n = 3–5 parallels for Western blot analysis. Statistical significance ** represents—*p* < 0.01 and *** represents—*p* < 0.001 compared to controls. Statistical significance + represents—*p* < 0.05, ++ represents—*p* < 0.01, and +++—*p* < 0.001 compared to PTX without GYY treatment. Statistical significance ## represents—*p* < 0.01 and ###—*p* < 0.001 compared to GYY4137. Full-length blots are presented in Supplemental Appendix A.

**Figure 2 biomolecules-14-00651-f002:**
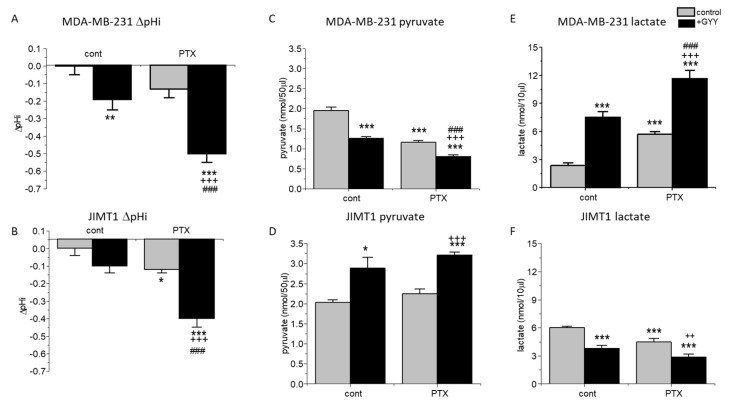
Intracellular pH, pyruvate, and lactate in MDA-MB-231 and JIMT1 cells after PTX and/or GYY4137 treatment. The intracellular pH (**A**,**B**) levels of pyruvate (**C**,**D**) and lactate (**E**,**F**) in MDA-MB-231 (**A**,**C**,**E**) and JIMT1 cells (**B**,**D**,**F**) were measured after treatment with paclitaxel (PTX; 20 nM), slow-releasing sulfide donor GYY4137 (GYY; 100 µM) and a combination of both PTX and GYY. We observed a significant decrease in intracellular pH after GYY treatment and after treatment with the combination of PTX and GYY in MDA-MB-231 cells (**A**). We observed a significant decrease in intracellular pH after PTX treatment and after treatment with the combination of PTX and GYY in JIMT1 (**B**). In MDA-MB-231 cells, a significant decrease in pyruvate production occurred in groups treated with GYY4137, PTX, and a combination of both compounds (**C**). In JIMT1 cells, a significant increase in pyruvate production after GYY4137 and its combination with PTX, but not PTX alone, occurred (**D**). A significant increase of lactate levels in MDA-MB-231 cells (**E**) and a significant decrease in JIMT1 cells (**F**) has been detected after the treatment with GYY, PTX, and both GYY and PTX. Results are displayed as mean ± S.E.M., and represent an average of the following cultivations for intracellular pH determination: n = 5 for MDA-MB-231, and n = 9 for JIMT1. Pyruvate and lactate levels were determined from four different cultivations each. Statistical significance * represents—*p* < 0.05, ** *p* < 0.01, and ***—*p* < 0.001 compared to untreated controls. Statistical significance ++ represents *p* < 0.01, and +++ *p* < 0.001 compared to PTX. Statistical significance ### represents—*p* < 0.001 compared to GYY.

**Figure 3 biomolecules-14-00651-f003:**
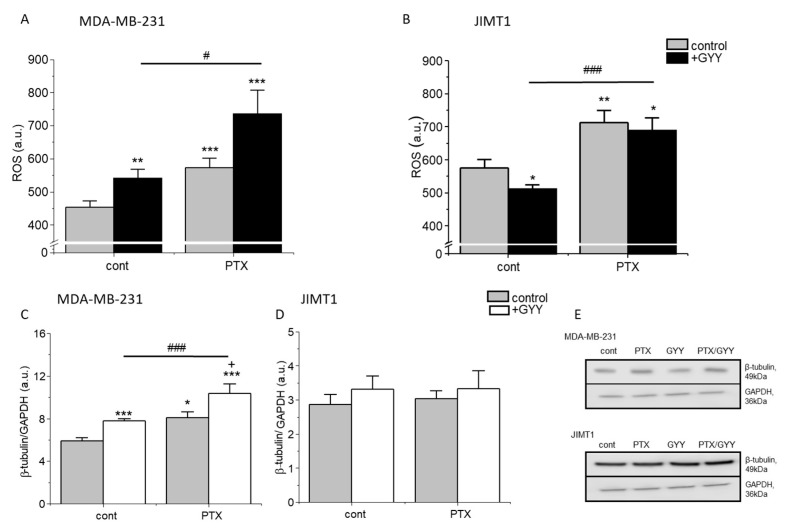
Production of ROS and β-tubulin in MDA-MB-231 and JIMT1 cells treated with PTX and/or GYY4137. We compared the production of reactive oxygen species (ROS; (**A**,**B**)) and β-tubulin in MDA-MB-231 cells (**C**,**E**) and JIMT1 cells (**D**,**E**) after treatment with paclitaxel (PTX; 20 nM), GYY4137 (GYY; 100 µM), or a combination of both. We observed significant ROS changes in all treated groups compared to the control untreated group in MDA-MB-231 (**A**) and in JIMT1 cells (**B**). Levels of β-tubulin were significantly increased in PTX and its combination with GYY4137 groups in MDA-MB-231 (**C**), but not in JIMT1 (**D**) cells. Typical gel images are enclosed (**E**). Results are displayed as mean ± S.E.M., and represent an average of six cultivations for MDA-MB-231 and four cultivations for JIMT1. Western blot analysis was evaluated from 3–5 gels. Statistical significance * represents—*p* < 0.05, ** represents—*p* < 0.01, and *** represents—*p* < 0.001 compared to untreated controls. Statistical significance + represents—*p* < 0.05 compared to PTX. Statistical significance between GYY4137 and PTX/GYY4137 represents #—*p* < 0.05, and ###—*p* < 0.001. Full-length blots are presented in Supplemental Appendix A.

**Figure 4 biomolecules-14-00651-f004:**
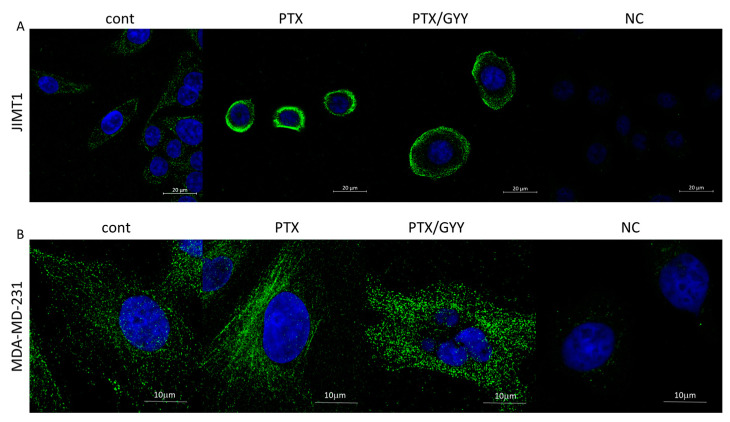
Immunofluorescence of β-tubulin on MDA-MB-231 cells and JIMT1 cells determined using a confocal microscope. In JIMT1 cell, rapid reorganization of the cytoskeleton (green) occurred due to paclitaxel (PTX; 20 nM) treatment, where the cytoskeleton seems to be condensed around the nucleus (**A**). In combined treatment of PTX and GYY4137 (GYY; 100 µM), the cytoskeleton is not so condensed, and is partially renewed (**A**). The rapid increase in β-tubulin filaments occurred after treatment with PTX alone, and filaments were disintegrated in combination with GYY4137 in MDA-MB-231 cells (**B**). Nuclei are stained by DAPI (blue). Images were taken in triplicates. NC—negative control, where the primary antibody was omitted.

**Figure 5 biomolecules-14-00651-f005:**
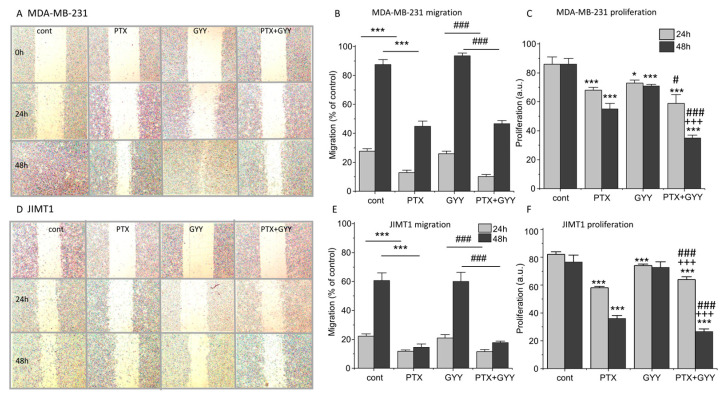
Migration and proliferation of MDA-MB-231 and JIMT1 cells after PTX and/or GYY4137 treatment. Migration of MDA-MB-231 cells (**A**,**B**) and JIMT1 cells (**D**,**E**) was evaluated after the treatment with paclitaxel (PTX; 20 nM), GYY4137 (GYY; 100 µM), and a combination of both these compounds using a wound healing assay. Migration was determined after 24 (light gray columns) and 48 (dark gray columns) hours. MDA-MB-231 cells migrate quicker than JIMT1 cells (**A**,**D**). Typical results of migration are shown for MDA-MB-231 cells (**A**) and JIMT1 cells (**D**). We observed significantly slower migration after treatment with PTX and a combination of PTX and GYY4137 in MDA-MB-231 (**A**,**B**) and JIMT1 (**D**,**E**) cells. Proliferation was lower after the PTX treatment in both cell lines, and the decrease was potentiated after combined treatment of PTX and GYY4137 (**C**,**F**). Results are displayed as mean ± S.E.M., and represent an average of six cultivations of MDA-MB-231 and JIMT1 for 24 h, four cultivations of MDA-MB-231, and five cultivations of JIMT1 for 48 h. Statistical significance * represents *p* < 0.05 and *** represents *p* < 0.001 compared to untreated controls. Statistical significance # represents *p* < 0.05 and ### represents—*p* < 0.001 compared to GYY4137. Statistical significance +++ represents *p* < 0.001 compared to PTX.

**Figure 6 biomolecules-14-00651-f006:**
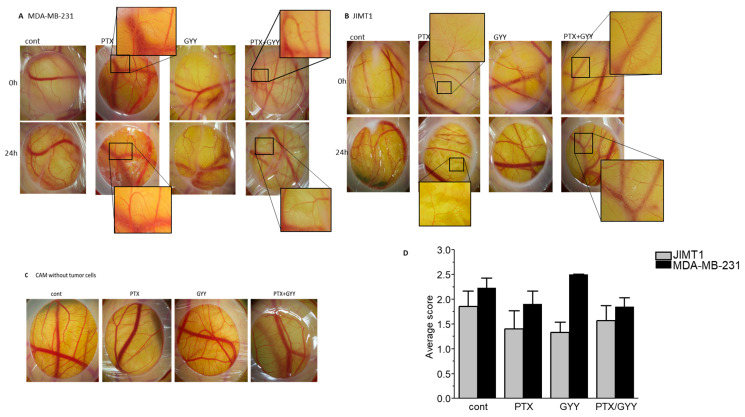
Determination of angiogenesis in MDA-MB-231 and JIMT1 cells after PTX and/or GYY4137 treatment. Effect of paclitaxel (PTX; 20 nM), GYY4137 (GYY; 100 µM), and PTX + GYY4137 treatment on angiogenesis of MDA-MB-231 (**A**) and JIMT1 (**B**) cell-line tumors on the CAM. CAMs without tumor cells are shown in (**C**). The classification score (**D**) was used to evaluate the effect of PTX, GYY4137, and PTX + GYY4137 treatment on MDA-MB-231 and JIMT1 tumor cell lines on CAM angiogenesis. cont—images of untreated tumors grafted onto the CAM. Insets show the magnified area, where differences in angiogenesis due to the treatment are clearly visible. In MDA-MB-231 cells, typical results of five embryos treated with GYY4137 and ten control embryos or embryos treated with PTX and/or PTX + GYY4137 are displayed. In JIMT1 cells, typical results of eight embryos treated with GYY4137 or PTX and nine control embryos or embryos treated with PTX + GYY4137 are displayed.

## Data Availability

The datasets used and/or analyzed during the current study are available from the corresponding author upon reasonable request.

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
