# Peer review of "Slow Sulfide Donor GYY4137 Increased the Sensitivity of Two Breast Cancer Cell Lines to Paclitaxel by Different Mechanisms"

_biomolecules, 2024, doi:10.3390/biom14060651_

Round 1

Reviewer 1 Report

Comments and Suggestions for Authors

Summary

The manuscript titled ‘Slow sulfide donor GYY4137 increased the sensitivity of two breast cancer cell lines to paclitaxel by different mechanisms’ focuses on the effect of GYY4137 on the MDA-MB-231 and JIMT1 cell line in the presence of paclitaxel. The authors use a variety of assay to show how GYY 4137 influences apoptosis and demonstrate that the compound promotes cell death in the 2 cell lines by different mechanisms. 

Strength

The manuscript is well written. There are a variety of assays performed to study apoptosis. 

Weakness

One of the major weakness is that while the authors show that the 2 cell lines respond differently to the sulfur donor, they have not specified what mechanism is involved. The discussion is vague on this aspect. Additionally, it is mentioned in most places that IP3R1 is not involved in cell death but then the conclusion states that the compound may use this protein in MDA-MB-231 apoptosis. This should be clarified and made consistent. 

The angiogenesis experiment, as shown in figure 6, is hard to understand from the perspective of the reader. The authors are recommended to provide a supporting graph based on their scoring system to complement the information obtained from the images. 

There are minor grammatical mistakes at different places. It would be good if better proof-reading was done.

I am not sure if this is an issue with the draft from the publisher or if this is how the authors provided the draft. The subscript in H2S does not look like a subscript.  

Comments on the Quality of English Language

Overall, the quality is good. Some minor errors are present that can be easily corrected by proof reading. 

Reviewer 2 Report

Comments and Suggestions for Authors

Chovancova et al. biomolecules-2961099" Slow sulfide donor GYY4137 increased the sensitivity of two breast cancer cell lines to paclitaxel by different mechanisms" is a valuable paper showing that a slow-releasing sulfide donor of H2S can potentiate the effect of paclitaxel in two breast cancer cell lines, MDA-MB-231 and JIMT1. Additionally, the authors showed that the mechanism of the effect in MDA-MB-231 and JIMT1 cells is probably different. However, some points are difficult for the reviewer to understand. The reviewer hopes that providing more information (described below) will improve the quality of this study. 

1. Page 6, lines 267-269 –the authors described that Treatment with GYY4137 alone was not toxic for either MDA-MB-231 (IC50-66 mmol/L) or for the JIMT1 cell line (IC50- 34.5 mmol/L). The reviewer could not find any data to support this finding. The reviewer believes that data should be presented.

2. Page 6, lines 277 – The authors described that necrosis in DTT- and TCEP-treated cells co-treated with PTX was much higher than that in other groups. There are no results in this study that show this. However, the reviewers believe that the results should be presented and discussed further.

3. Results section: The purpose of each experiment is not stated in the results section; it appears to be stated in the discussion section. Reviewers believe that for the reader to understand why the experiment was performed, the paper needs to be constructed by stating the purpose of each experiment.

Reviewer 3 Report

Comments and Suggestions for Authors

The article titled " Slow sulfide donor GYY4137 increased the sensitivity of two breast cancer cell lines to paclitaxel by different mechanisms" by Barbora Chovancova et al. provides valuable insights. However, there are several areas that require attention:

1. The authors must perform a comprehensive evaluation of cytotoxicity with dose and time-dependent manners.

2. Authors should investigate the colony formation assay and proliferation (Ki67, PCNA etc.) to determine the long-term effects of the GYY4137 on breast cancer cells

3. The authors did not perform any in vivo experiments in the present study and did not obtain consistent outcomes regarding the Slow sulfide donor GYY4137 increased the sensitivity to paclitaxel on TNBC-xenograft animal models.

4. This work is still preliminary and needs to explore the underlying downstream pathways. Further research is needed to evaluate the effects of these GYY4137 on other relevant breast cancer phenotypes and signaling pathways involved in breast cancer development and progression.

5. The authors should evaluate the potential side effects or toxicity of GYY4137, which is important for determining their safety and suitability for animal studies or clinical use.

6. Authors may include invasion assay (using matrigel coated) and EMT markers using WB

Comments on the Quality of English Language

Moderate editing of English language required

Reviewer 4 Report

Comments and Suggestions for Authors

The article is interesting, but not very clear and understandable. The methodology is described in detail, but the results are not well presented. The introduction and discussion require modification.

I have many comments on the manuscript and it requires a major revision

1.      I don't understand the statements [see review 1], [for review see 13]

2.      Unnecessarily duplicate information, e.g. about cell lines in the introduction and materials and methods

3.      In turn, the use of DTT and TCEP as controls is not explained.

4.      Where in Figure 1 this statement is shown ‘Treatment with GYY4137 alone was not toxic for either MDA-MB-231 (IC50-66 mmol/L) or for the JIMT1 cell line (IC50- 34.5 mmol/L)’.

5.      These two sentences are mutually exclusive to me: Paclitaxel IC50 values (PTX) and the combination of PTX and GYY4137 (GYY) are significantly lower in JIMT1 compared to MDA-MB-231 cells. Based on the IC50 values of JIMT1 and MDA-MB-231, the PTX concentration in all other experiments was set to 20 nM.

6.      In Figure 1B, the wording untreated and GYY itself is misleading. The description indicates otherwise. 'Different concentrations of PTX (2 nM, 20 nM and 200 nM) were used with/without GYY4137 (100 µM) to measure apoptosis (B)'

7.      Figure 1 should be rebuilt. Analyzing this number takes a lot of time and not all results are presented. The figure should be prepared in such a way that it is not necessary to read half of the article to draw conclusions.

8.      Consistent descriptions in figures e.g. GYY, GYY4137, GYY437

9.      Poor quality Fig 4A

10.   No bar scale in 4B

11.   I can't read the scale in Figure 4A

12.   ‘Changes in migration due to PTX and PTX/GYY4137 treatment were determined by a wound healing assay. The migration of the MDA-MB-231 cells was more rapid than the migration of the JIMT1 cells (Figure 5). The size of the wound was determined immediately after the scratch and after 24 and 48 hours (Figure 5A,C). In both, MDA-MB-231 (Figure 5A,B) and JIMT1 (Figure 5C,D) cells the migration was significantly slower in groups treated with PTX or a combination of PTX and GYY4137, compared to GYY4137 alone. Migration in the presence of GYY4137 was similar to control, untreated cells (Figure 5).’ This description is correct, but why is there no information about the lack of differences between PTX and PTX and GYY4137.

13.   The discussion is a significant repetition of the introduction and results, so I propose to shorten it significantly.

Reviewer 5 Report

Comments and Suggestions for Authors

The manuscript "Slow sulfide donor GYY4137 increased the sensitivity of two breast cancer cell lines to paclitaxel by different mechanisms" is dedicated to the study of the mechanism of action of the compound GYY4137. Paclitaxel was used as a comparator. The work was performed on two different breast cancer cell lines: MDA-MB-231 and JIMT1. A near-record number of biological studies were performed in the article: migration detection, immunofluorescence, determination of lactate and pyruvate, determination of apoptosis, measurement of oxidative stress, and angiogenesis studies. The work is performed at a very high level and certainly deserves to be published in the scientific journal Biomolecules. However, I did not find the flow cytometry rafts in the SI file. It is very important to cite flow cytometry results in histograms and scatter plots in the SI file. The authors use the abbreviation IC50 in the article, but it would be much more correct to use CC50 since the cellular targeting for the GYY4137 compound is not yet known. Also, paclitaxel has not been sufficiently studied, although one of its mechanisms is inhibition of tubulin synthesis. Therefore, I strongly recommend replacing IC50 with CC50.    

Round 2

Reviewer 2 Report

Comments and Suggestions for Authors

The second revised paper seems to be improved well and should be worth publishing in this journal.

Reviewer 3 Report

Comments and Suggestions for Authors

Accept in present form

Comments on the Quality of English Language

Minor editing of English language required

Reviewer 4 Report

Comments and Suggestions for Authors

The manuscript has been revised according to the suggestions and comments.